# The GPR18 Agonist PSB-KD-107 Exerts Endothelium-Dependent Vasorelaxant Effects

**DOI:** 10.3390/ph14080799

**Published:** 2021-08-13

**Authors:** Magdalena Kotańska, Monika Kubacka, Marek Bednarski, Noemi Nicosia, Małgorzata Szafarz, Wojciech Jawień, Christa E. Müller, Katarzyna Kieć-Kononowicz

**Affiliations:** 1Department of Pharmacological Screening, Medical College, Jagiellonian University, Medyczna 9, PL 30-688 Cracow, Poland; marek.bednarski@uj.edu.pl (M.B.); noemi.nicosia92@gmail.com (N.N.); 2Department of Pharmacodynamics, Medical College, Jagiellonian University, Medyczna 9, PL 30-688 Cracow, Poland; 3Foundation “Prof. Antonio Imbesi”, University of Messina, Piazza Pugliatti 1, 98122 Messina, Italy; 4Department of Chemical, Biological, Pharmaceutical and Environmental Sciences, University of Messina, Viale Palatucci, 98168 Messina, Italy; 5Department of Pharmacokinetics and Physical Pharmacy, Medical College, Jagiellonian University, Medyczna 9, PL 30-688 Cracow, Poland; malgorzata.szafarz@uj.edu.pl; 6Department of Pharmaceutical Biophysics, Faculty of Pharmacy, Medical College, Jagiellonian University, Medyczna 9, PL 30-688 Cracow, Poland; wojciech.jawien@uj.edu.pl; 7Pharma Center Bonn, Pharmaceutical & Medicinal Chemistry, Pharmaceutical Institute, University of Bonn, An der Immenburg 4, D-53121 Bonn, Germany; christa.mueller@uni-bonn.de; 8Department of Technology and Biotechnology of Drugs, Faculty of Pharmacy, Medical College, Jagiellonian University, Medyczna 9, PL 30-688 Cracow, Poland; katarzyna.kiec-kononowicz@uj.edu.pl

**Keywords:** agonist, cannabinoid, GPCR, GPR18, orphan receptor, vasodilation

## Abstract

GPR18 is an orphan GPCR that is activated by the cannabinoid tetrahydrocannabinol (THC). Emerging evidence indicates its involvement in the control of cardiovascular functions, including heart rate, contractility, vascular tone, as well as blood pressure. Therefore, we investigated the effects of selective GPR18 receptor ligands, namely PSB-KD-107 (agonist) and PSB-CB-92 (antagonist), on blood pressure, electrocardiogram (ECG), and vascular dilatation in vitro and in vivo, as well as their anti-oxidative potential in in vitro ferric reducing antioxidant power (FRAP) and 2,2-diphenyl-1-picryl-hydrazyl-hydrate free radical (DPPH) assays. Our results clearly show that PSB-KD-107 dilates blood vessels. This effect is related to its activation of GPR18 as it can be blocked by the GPR18 antagonist PSB-CB-92. Moreover, our finding confirms the presence of GPR18 in blood vessels. The mechanism of the vasorelaxant activity of PSB-KD-107 is mainly related to endothelial nitric oxide generation; however, we cannot exclude additional nitric oxide-independent mechanisms or a direct influence on K^+^ channels. PSB-KD-107 may affect blood pressure and heart function after a single administration; however, this effect was no longer observed after repeated administrations once daily for eight days. PSB-KD-107 does not affect platelet aggregation—an important feature considering the safety of its administration. PSB-KD-107 also shows a significant anti-oxidant effect and further studies of its antioxidant activity in vivo are justified.

## 1. Introduction

A growing number of studies investigating the endocannabinoid system has highlighted its involvement in various physiological or pathological processes. This evidence may offer valuable insights, eventually leading to the development of therapeutic drugs for a variety of conditions ranging from mood disorders, obesity, metabolic syndrome, as well as neurodegenerative and cardiovascular diseases [1,2,3,4].

The endocannabinoid system comprises neuromodulatory molecules, commonly known as endocannabinoids, whose activity is mediated through cannabinoid receptors and enzymes involved in the biosynthesis and degradation of endocannabinoids [5]. Anandamide (AEA) and 2-arachidonoylglycerol (2-AG) were the first isolated endogenous cannabinoid receptor ligands. They are derivatives of omega-6-polyunsaturated fatty acids and are widespread in the human brain. Further endocannabinoids, including 2-arachidonoylglyceryl ether (noladin ether), O-arachidonoylethanolamine (virodhamine), and N-arachidonoyl-dopamine have been identified [6].

Endocannabinoids exhibit a high affinity for CB_1_ and CB_2_, the archetypal cannabinoid receptors belonging to the family of G protein-coupled receptors (GPCR), which are widely distributed throughout the body. CB_1_ receptors are mostly expressed in the central nervous system (CNS), where they mediate central cannabinoid effects. Low concentrations of CB_1_ receptors have also been detected peripherally in the spleen, testes, vascular endothelium, retina, enteric nervous system, and adipocytes [7]. CB_2_ receptors are expressed in cells of the immune system, including microglia and astrocytes in the CNS. Therefore, they are associated with anti-inflammatory and immunosuppressant effects [8]. They are involved in different pathological conditions such as neurodegeneration, neuropathic pain, multiple sclerosis, and amyotrophic lateral sclerosis [7].

Recently, more and more studies emphasize the role of orphan GPCR such as GPR18, GPR55, and GPR119, as molecular targets for both endogenous cannabinoid ligands and abnormal cannabinoids (Abn CNB) [9]. GPR18 is clearly activated by the natural cannabinoid Δ^9^-tetrahydrocannabinol (THC). Endogenous agonists have been postulated, including AEA, its metabolite N-arachidonylglycine (NAGly), and resolvin D2, but these results are controversially discussed in the literature [10,11]. THC is a non-selective ligand for GPR18, it is much more potent as a partial CB_1_ and CB_2_ receptor agonist, and it additionally blocks GPR55, an orphan receptor that is activated by lysophosphatidylinositol. GPR119 is activated by N-oleoylethanolamine and 2-oleoylglycerol, monounsaturated fatty acid analogs of AEA and 2-AG, and N-oleoylethanolamine [12]. GPR18 has been classified as the third cannabinoid receptor following CB_1_ and CB_2_, although further studies are needed to discover and explore its native agonist and to elucidate its (patho)physiological roles and its potential as a novel therapeutic target [13]. 

Our latest studies have allowed us to identify new tool compounds for GPR18, which are selective, versus other cannabinoid receptors, including agonists [11] and antagonists [12]. Schoeder et al. (2020) discovered the tricyclic xanthines PSB-KD-107 and PSB-KD477 as the first non-lipid-like agonists for GPR18. Moreover, these compounds displayed higher potency and efficacy than THC [11]. In addition, PSB-KK-1415 has recently been developed as a novel potent and selective cannabinoid receptors with increased potency agonist [4,14]. Derivatives of imidazothiazinones, namely PSB-CB5, PSB-CB27, and PSB-CB-92 are the most potent synthetic GPR18 antagonists developed to date [12,15].

Although it has not been clearly demonstrated so far if GPR18 is expressed in blood vessels, emerging findings provide evidence of its involvement in the control of cardiovascular functions, including heart rate, contractility, vascular tone, and blood pressure [10,16]. GPR18 was originally described as the endothelial cannabinoid receptor because its activation by anandamide or by the synthetic cannabinoid Abn CNB caused vasodilation and hypotension [17,18]. In the studies published so far on this topic, the ligands used are not selective versus other cannabinoid receptors [16,19,20]; moreover, their activation of GPR18 could not be confirmed by some laboratories [11,21]. Therefore, in the current study, we tested the influence of selective GPR18 ligands, the agonist PSB-KD-107, and the antagonist PSB-CB-92 on blood pressure, ECG, and vascular dilatation in in vitro and in in vivo research. These compounds are expected to be useful research tools for elucidating the mechanisms of action of GPR18 in blood vessels. Additionally, since the antioxidant balance is of great importance in the treatment of cardiovascular disorders, we complemented our studies with in vitro antioxidant assays. 

## 2. Results

### 2.1. Effect Induced by PSB-KD-107 and PSB-CB-92 in Rat Aorta Precontracted with Phenylephrine

Cumulative concentrations of PSB-KD-107 (0.3–100 µM) produced concentration-dependent relaxation in phenylephrine-precontracted endothelium-intact aortic rings, with a pIC_50_ value of 5.22 ± 0.020 (Figure 1, Table 1). On the other hand, the antagonist PSB-CB-92 did not show any significant vasodilating effect in a concentration range of 0.1–30 μM. The maximal effect observed at a concentration of 30 μM was approximately 20.21%. The compound PSB-CB-92 was not tested at the higher concentrations due to its precipitation in the Krebs-Henseleit solution.

To determine whether the observed vasodilatory property of PSB-KD-107 was due to its activity at GPR18, the effect of PSB-KD-107 on isolated rat aorta stimulated with phenylephrine was examined after its prior incubation with PSB-CB-92 (10 µM and 20 µM), the GPR18 antagonist. Prior incubation with PSB-CB-92 reduced the vasodilatory effect of PSB-KD-107, shifting the dose-response curve to the right (Figure 2). The pIC_50_ value for PSB-KD-107 in the presence of PSB-CB-92 (10 µM) was 4.884 ± 0.020, and it was significantly different from the pIC_50_ obtained for PSB-KD-107 without pretreatment with PSB-CB-92 (*p* < 0.0001). Pretreatment with a higher concentration of PSB-CB-92 (20 µM) resulted in a further shift of the dose-response curve to the right, suggesting that the vasodilatory effect of PSB-KD-107 is, at least partially, due to its action at GPR18. The pIC_50_ value for PSB-KD-107 in the presence of PSB-CB-92 (20 µM) was 4.735 ± 0.023 (*p* < 0.0001), (Figure 2, Table 1).

To test whether the relaxant effect of PSB-KD-107 is endothelium-dependent, its vasodilatory properties were also tested in the absence of endothelium. Denudation of functional endothelium decreased the relaxant effect (Figure 3, Table 1), resulting in the pIC_50_ value of 4.904 ± 0.023.

To evaluate the involvement of endothelium-dependent effects in the PSB-KD-107 induced vasorelaxation, the effects of N_ω_-Nitro-L-arginine methyl ester hydrochloride (L-NAME, 100 µM), a non-selective nitric oxide synthase (NOS) inhibitor, were tested. Figure 3 and Table 1 show that pre-incubation of endothelium-intact aortic rings with L-NAME markedly attenuated the vasorelaxation induced by PSB-KD-107. The pIC_50_ value after such a pretreatment was in the same range as the pIC_50_ value observed in endothelium-denuded aortic rings. Preincubation with L-NAME reduced the pIC_50_ value from 5.217 ± 0.020 to 4.873 ± 0.015 (*p* < 0.0001).

To investigate the involvement of cyclooxygenase (COX) products in the PSB-KD-107-induced relaxation, the effect of indomethacin (10 µM), a COX inhibitor, was studied. Indomethacin did not influence the vasorelaxant activity of PSB-KD-107, which indicates that COX products were not involved in its mechanism of action (Figure 3, Table 1).

To assess the possible involvement of K^+^ channels in the PSB-KD-107 induced relaxation, the aortic rings were preincubated with tetraethylammonium (TEA, 1 mM), a non-selective inhibitor of K^+^ channels. TEA reduced PSB-KD-107 induced vasorelaxation, with a pIC_50_ value decreasing from 5.217 ± 0.020 to 4.940 ± 0.030 (*p* < 0.0001), and the maximal vasodilatory effect at 100 µM from 100% to 83.5 ± 1.5%

### 2.2. Effect of PSB-KD-107 on Guinea-Pig Ileum Contraction Induced by Carbachol

The effect of PSB-KD-107 on carbachol-induced contractions was measured in guinea-pig ileum. PSB-KD-107 at a concentration of 3 µM displayed no significant ability to block the contraction induced by carbachol (CCH). However, at a concentration of 10 µM, it depressed the maximal CCH response to 80.67 ± 4.06%, without shifting the dose-response curve, which suggests a non-competitive antagonism (Figure 4). 

### 2.3. Effect of PSB-KD-107 and PSB-CB-92 on Platelet Aggregation

To assess the effects of the GPR18 ligands on platelet aggregation, freshly isolated rat whole blood was incubated with test compounds or vehicle (0.1% DMSO), and the aggregation responses were evaluated with a whole blood aggregometer by measuring the change of impedance. Platelet aggregation was induced by collagen. PSB-KD-107 did not influence platelet aggregation even at a high concentration of 100 µM. Similar observations were found for PSB-CB-92; however, the antagonist was tested only at the concentration of 10 µM due to its limited solubility. The results are presented in Figure 5. 

### 2.4. Antioxidant Activity of PSB-KD-107 and PSB-CB-92

#### 2.4.1. Ferric Reducing Antioxidant Power (FRAP) Assay

Agonist PSB-KD-107 was tested in the FRAP test; the compound produced Fe^2+^ ions in direct proportion to its concentration. Within the range of tested concentrations, i.e., 100–1000 µM, every 100 micromoles of the compound reduced 131 micromoles of the Fe^3+^ (Figure 6).

The reference compound, ascorbic acid, reduced 190 micromoles of the Fe^3+^ for every 100 micromoles of the compound. Thus, compound PSB-KD-107 demonstrated antioxidant activity at a level of 60–80% of that of ascorbic acid.

Compound PSB-CB-92, even at the concentration of 1000 µM, showed no antioxidant activity in this test.

#### 2.4.2. 2,2-Diphenyl-1-picryl-hydrazyl-hydrate Free Radical (DPPH) Assay

Both compounds, tested at two concentrations (100 and 1000 µM), did not cause a decrease in the absorbance, which indicates a lack of scavenging of the DPPH free radical and the antioxidant effect. The reference compound (ascorbic acid) in this test showed antioxidant activity in the concentration range of 100–1000 µM (Figure 7).

### 2.5. Influence of PSB-KD-107 and PSB-CB-92 on Blood Pressure in Normotensive Rats

After a single administration of PSB-KD-107 at a dose of 10 mg/kg intraperitoneally (i.p.) to normotensive rats, significantly decreased the systolic (by approx. 11–12 mmHg) and diastolic (by approx. 13 mmHg) blood pressure, 40–60 min after its administration (Figure 8). At lower doses, PSB-KD-107 did not significantly influence blood pressure (data not shown).

Compound PSB-KD-107 after multiple administrations (eight times, once daily) at a dose of 10 mg/kg i.p. had no statistically significant effect on blood pressure, compared to the control group receiving vehicle (1% Tween; 1 mL/kg i.p.), (Figure 9).

Compound PSB-CB-92 after both single- and multiple (eight times, once daily) administrations at a dose of 10 mg/kg b.w. i.p. did not affect blood pressure. There were no significant differences in comparison with the blood pressure values of control animals receiving vehicles (Figure 10 and Figure 11).

### 2.6. The Effects of PSB-KD-107 and PSB-CB-92 on the Normal Electrocardiogram (ECG)

Compound PSB-KD-107 at a dose of 10 mg/kg i.p. did not significantly affect the ECG recording in terms of duration of the PQ interval, the QRS complex, and the QT interval. Only 40 min after administration, PSB-KD-107 caused a decrease in heart rate by about 30 beats/min, which was accompanied by a significant prolongation of the QT interval. As the duration of the QT interval depends on the heart rate, a corrected QT interval (QTc) was also calculated, and this parameter showed no abnormalities (Table 2).

Compound PSB-CB-92 administered i.p. at a dose of 10 mg/kg did not significantly affect the ECG recording in terms of the heart rate, duration of the PQ interval, QRS complex, or QT interval. It only decreased the number of cardiac beats per minute in the 5th min of observation, and later the heart rate returned to the baseline (Table 2). 

Both compounds, PSB-KD-107 and PSB-CB-92, administered once daily for 8 days did not significantly affect the heart rate (Figure 12A) or the other components of the ECG recording: i.e., PQ interval (Figure 12B), QRS complex (Figure 12C), or QT interval (Figure 12D).

## 3. Discussion

Having GPR18 ligands that are selective versus other cannabinoid receptors, we investigated their activity in the cardiovascular system. Due to the activity profile of PSB-KD-107 and PSB-CB-92, our studies differ significantly from those published so far on this topic and are very valuable because we could use, for the first time, specific ligands for GPR18 to perform these experiments.

In our study, we found that PSB-KD-107—selective agonist of GPR18—exerted vascular activity and was able to produce total relaxation of phenylephrine-pre-contracted rat aorta. On the other hand, PSB-CB-92, an antagonist of GPR18, did not show significant vasorelaxant effects, and the vasorelaxation did not exceed 20.21% at the highest concentration tested.

To evaluate if the vasodilatory properties of PSB-KD-107 result from GPR18 activation, its effect was assessed after pre-incubation of blood vessels with the selective GPR18 antagonist PSB-CB-92. The vasorelaxation produced by PSB-KD-107 was in fact significantly reduced in the presence of the antagonist PSB-CB-92, and the concentration-response curve was shifted to the right in a parallel and dose-dependent manner. This demonstrates that PSB-CB-92 blocks the effect of PSB-KD-107, probably due to a competitive mechanism of inhibition at a common target site, which is GPR18. The obtained results clearly show that stimulation of GPR18 results in vasodilation, while the simultaneous administration of agonist and antagonist of this receptor reduces this effect. The result of this experiment also indicates, although indirectly, the presence of GPR18 in blood vessels.

Comparison of the relaxation induced by PSB-KD-107 in intact endothelium and endothelium-denuded rat aorta showed that the endothelium denudation partially, but significantly, diminished PSB-KD-107 vasorelaxant activity. This suggests that functional endothelium is involved in the mechanism of this action.

Upon stimulation (with shear stress or agonists), the endothelial cells release vasorelaxant mediators such as NO, prostacyclin (PGI_2_), and endothelium-derived hyperpolarizing factor (EDHF). In rat aorta, NO is the main vasodilator produced in endothelial cells, whereas in resistance arteries, the contribution of EDHF to endothelium-dependent relaxations prevails [22]. Hence, we mainly explored the endothelial NO and PGI_2_ pathways. The vasorelaxant activity of PSB-KD-107 was reduced by the NOS inhibitor L-NAME to the level observed in the endothelium-denuded aorta. This clearly indicates that endothelial NO generation is involved in the PSB-KD-107-induced vasorelaxant effect. Contrary, the PGI_2_ pathway probably is not involved in the endothelium-dependent component of the vasorelaxant effect of PSB-KD-107, as indomethacin—the COX activity and prostacyclin production inhibitor, did not affect relaxation induced by PSB-KD-107. A similar observation was reported by Al. Suleimani et al. [19]. They found that the vasorelaxation produced by NAGly, an endogenous lipid that was previously reported to activate GPR18, exerts endothelium-dependent vasorelaxant activity, sensitive to blockade with L-NAME, but not to indomethacin. However, NAGly is controversially discussed as a GPR18 agonist, and the authors did not clearly demonstrate the role of GPR18 in the relaxant effect of NAGly due to the previous lack of selective GPR18 antagonists. In our study, we showed that GPR18 is involved in the vasorelaxant effect of the GPR18 agonist PSB-KD-107 as the selective GPR18 antagonist PSB-CB-92 induced a rightward shift of the agonist’s dose-response curve.

GPR18 was reported to be a Gi/G0-protein-coupled receptor and might lead to a reduction in cAMP levels and activation of the PI3K/Akt and ERK1/2 pathways [16], although this has been disputed [11,21]. As the PI3K/Akt-ERK1/2 pathway activates eNOS and NO generation, it may explain the vasodilation and hypotensive effect observed after GPR18 activation. 

The vasodilation induced by PSB-KD-107 was attenuated by TEA, a non-selective inhibitor of K^+^ channels, which indicates the participation of the K^+^ channels in the PSB-KD-107 vasorelaxant effect. K^+^ channels significantly contribute to the regulation of the vascular smooth muscle tone. The opening of K^+^ channels and efflux of K^+^ causes membrane hyperpolarization, leading to the closure of voltage-operated Ca^2+^ channels, reduction of Ca^2+^ influx, and vasodilation [23]. It is known that by increasing the cGMP level in vascular smooth muscle cells, NO also indirectly activates K^+^ channels through protein kinase G (PKG) [23]. Therefore, the vasorelaxant effect of PSB-KD-107 may be associated with the NO-cGMP-K^+^ channels pathway, as both L-NAME and TEA reduced its vasodilatory properties. A similar mechanism of action was previously observed for NAGly. Parmar et al. showed that NAGly stimulated the endothelial release of NO, which in turn activated BK(Ca) in the smooth muscle. In addition, NAGly might also activate BK(Ca) through NO-independent mechanisms [24]. Similarly, we cannot exclude the NO-independent or direct effect of PSB-KD-107 on K^+^ channels.

Our research also showed that endothelium denudation or pretreatment with L-NAME or TEA only partially impaired the vasorelaxant effect of PSB-KD-107, which means that mechanisms other than action through NO or K^+^ channels may be involved. In the experiment evaluating the effect of PSB-KD-107 on the CCH-induced contraction in guinea-pig ileum, PSB-KD-107 at high concentration depressed the maximal effect of CCH without shifting the dose-response curve, indicating a non-competitive antagonism. Reduction of the maximal response to CCH suggests an interaction between PSB-KD-107 and another receptor/channel system present in the guinea-pig ileum (i.e., Ca^2+^, K^+^ channels), which limits the maximal effect of CCH [25]. These unknown mechanisms may also be involved in the vasorelaxant effect of PSB-KD-107.

The effect of GPR18 receptor ligands on blood pressure is of great interest, especially due to the complex and unclear mechanism by which cannabinoids cause vasodilation and affect cardiac function [16,19,24,26]. Cannabinoid CB_1_ receptor activation modulates cardiovascular function via central and peripheral mechanisms. Direct activation of cardiac CB_1_ receptors exerts negative chronotropic effects, while activation of endothelial CB_1_ receptors results in vasodilation. Central CB_1_ regulation of blood pressure depends on the place of CB_1_ agonist microinjection; administration of cannabinoids into the periaqueductal gray (PAG) and rostral ventrolateral medulla (RVLM) triggers a sympathoexcitatory response, while administration in the nucleus tractus solitarii (NTS) is known to improve baroreflex sensitivity and facilitates inhibition of pressor responses [26]. PSB-KD-107 has been shown to be highly selective for GPR18 over GPR55, CB_1,_ and CB_2_ receptors. In radioligand binding studies at CB receptors, PSB-KD-107 tested at a high concentration of 10 μM showed no significant displacement of the CB receptor-specific radioligand at either CB_1_ or CB_2_ receptors. It also did not show any increase in radioligand binding that could have been indicative of positive allosteric CB receptor modulation [11]. On that basis, we can assume that the cardiovascular effects of PSB-KD-107 are CB_1_ receptors-independent. After a single administration to normotensive rats, PSB-KD-107 exerted hypotensive activity accompanied with bradycardia. This is consistent with the study presented by Penumarti and Abdel-Rahman [10], who demonstrated that NAGly acting through GPR18 in the RVLM was able to lower both blood pressure and heart rate. The observed cardiovascular effects of PSB-KD-107 may be in part attributed to its vasorelaxant properties and the activation of endothelial GPR18 receptors. GPR18 has been localized in RVLM and their activation, contrary to the CB_1_ receptor activation in RVLM [16], mediates a reduction in blood pressure [10] and a mild reduction in heart rate. The lack of expected reflex tachycardia in response to the vasodilatory effect of PSB-KD-107 may be explained by the action of PSB-KD-107 on GPR18 in RVLM and reduction in sympathetic activity. It is possible that PSB-KD-107 exerts both peripheral as well as centrally-mediated effects on blood pressure and heart rate. According to Lipinski’s “Rule of Five”, it is supposed to have good permeability. As for the good CNS penetration, PSB-KD-107 meets 3 out of 4 criteria (molecular weight ≤ 400; LogP ≤ 5; hydrogen bond donors ≤ 3) [27]. Thus, based on the theoretical assumptions, it is highly likely that PSB-KD-107 is able to cross the blood-brain barrier. However, future studies are necessary to verify this hypothesis.

Matouk et al. showed that chronic GPR18 activation, with its agonist abnormal cannabidiol (abn-CBD), produced hypotension, suppressed cardiac sympathetic dominance, and improved left ventricular function [16]. These effects were claimed to be due to a GPR18 mediated increase in NOS and NO expression and reduction of myocardial reactive oxygen species (ROS) production. In the studies using abn-CBD as a GPR18 agonist, no change in heart rate was seen. Interestingly we observed neither marked hypotensive activity nor bradycardia after semi-chronic administration of the selective GPR18 agonist PSB-KD-107. Of note, PSB-KD-107 is a weak antagonist of adenosine A_2A_ receptors [28], and this property may weaken its hypotensive activity since adenosine A_2A_ receptor antagonists have been found to increase arterial pressure, while adenosine via adenosine A_2A_ receptors are strongly hypotensive [29].

As blood platelets also express adenosine A_2A_ receptors, in the subsequent experiment, we aimed to assess the influence of PSB-KD-107 on platelet aggregation. Activation of platelet adenosine A_2A_ receptors leads to an enhancement of intracellular cAMP levels and consequently causes the inhibition of platelet activation and aggregation [30]. PSB-KD-107 then, an adenosine A_2A_ receptor antagonist, could possibly enhance platelet aggregation. However, it should be emphasized that PSB-KD-107 tested even at very high concentrations did not potentiate platelet aggregation induced by collagen.

Taking into account pharmacological safety, we also evaluated the effect of PSB-KD-107 on ECG recording after both single and semi-chronic administration. PSB-KD-107 caused no significant effect on the PQ interval, which suggests that it does not affect the atrio-ventricular conduction time. The results of ECG studies in rats also showed that PSB-KD-107 did not prolong the QT interval or alter the QRS complex, suggesting that it did not affect ventricular depolarization and repolarization. We also performed hemodynamic studies for PSB-CB-92, an antagonist of GPR18. PSB-CB-92 did not significantly influence blood pressure or affect ECG patterns.

Some cannabinoid receptor ligands, such as cannabidiol (CDB) or cannabigerol, have significant antioxidant activity [31,32,33,34,35]. Because of this property, they prevent the negative effects of oxidative stress, which is the cause of many disorders or disease complications [36,37,38]. Such an activity is very beneficial for many indications, such as neurodegenerative or cardiovascular diseases or, for example, diabetes. Antioxidant activity increases the potential and effectiveness of treatment, and the ligands that are also antioxidants are prioritized at the early stages of drug development. Therefore, we assessed the new GPR18 ligands’ antioxidant properties. We emphasize that this is one of the first published pharmacological studies related to these specific ligands of GPR18.

Penumarti and colleagues reported that activation of the GPR18 by abn-CBD leads to a reduction in ROS levels in the core tissue [10]. McPartland and colleagues reported that CBD can induce ROS in cancer cells and reduce the amount of ROS in healthy cells stimulated by factors that induce their formation [39]. In our in vitro tests, PSB-KD-107 did not reduce the amount of free radicals (DPPH test) but based on the examples presented above, we cannot exclude that (in vivo) it would have such activity related to the stimulation of GPR18. Therefore, future in vivo experiments verifying the effect of PSB-KD-107 on the oxidation-reduction balance and allowing for the determination of specific mechanisms of its antioxidant activity in the cardiovascular system is warranted. It also must be pointed out that our results regarding the role of GPR18 and their ligands in blood vessels are preliminary and future studies employing different approaches are necessary to confirm the obtained results. In conclusion, the current study sheds light on the effects of GPR18 activation and blockades in the cardiovascular system. 

## 4. Materials and Methods

### 4.1. Animals

The experiments were carried out on male Wistar rats (Krf:(WI) WU), 200–250 g) obtained from an accredited animal house at the Faculty of Pharmacy, Jagiellonian University Medical College, Krakow, Poland, and male Outbred CV guinea pigs (250–300 g) were purchased from a licensed breeder (Ilkowice 41, 32–218 Słaboszów, Poland). The total number of animals used in the study was 90 rats and 3 guinea pigs. The animals were housed in constant temperature (22–24 °C) and humidity (40–60%) facilities, exposed to 12:12 h light/dark cycles, and were maintained on a standard pellet diet with tap water available *ad libitum*. Animals were selected randomly, and all the experiments were performed between 9:00–14:00. All procedures were conducted in accordance with the ARRIVE guidelines and with the guidelines of the National Institutes of Health Animal Care and Use Committee and approved by the Local Ethics Committee of Jagiellonian University in Kraków (resolution no. 150/2018). 

### 4.2. Drugs and Chemicals

Compounds PSB-CB-92 and PSB-KD-107 were synthesized at the Department of Technology and Biotechnology of Drugs, Pharmaceutical Faculty, Jagiellonian University, and their GPR18 activity was discovered at the Department of Pharmaceutical & Medicinal Chemistry, Pharmaceutical Institute University of Bonn [11,12,15,28].

The materials used such as acetylcholine hydrochloride, carbamoylcholine chloride (carbachol, CCH), N_ω_-Nitro-L-arginine methyl ester hydrochloride (L-NAME), phenylephrine hydrochloride, and tetraethylammonium chloride (TEA) were purchased from the Sigma-Aldrich (Darmstadt, Germany), thiopental sodium was obtained from the Biochemie GmbH (Kundl, Austria). Other chemicals used were obtained from POCH (Polish Chemical Reagents, Gliwice, Poland). 

PSB-CB-92 and PSB-KD-107 were dissolved in DMSO immediately before use for in vitro studies or suspended in 1% Tween for in vivo studies.

### 4.3. In Vitro Studies

#### 4.3.1. Influence of PSB-KD-107 and PSB-CB-92 on Rat Aorta Precontracted with Phenylephrine

Rats were anesthetized with thiopental sodium (75 mg/kg i.p.), the thoracic aortas were dissected, placed in a Krebs-Henseleit solution (NaCl 119 mM, KCl 4.7 mM, CaCl_2_ 1.9 mM, MgSO_4_ 1.2 mM, KH_2_PO_4_ 1.2 mM, NaHCO_3_ 25 mM, glucose 11 mM, EDTA 0.05 mM) and cleaned of surrounding fat tissues. Two stainless-steel stirrups were inserted through the lumen of each arterial segment: one stirrup was attached to the bottom of the chamber and the other to an isometric FDT10-A force-displacement transducer (BIOPAC Systems, Inc., COMMAT Ltd., Ankara, Turkey). The rings were placed in a 30 mL organ chamber that contained Krebs-Henseleit solution oxygenated with 95%O_2_/5%CO_2_ and maintained at 37 °C. In some aortic rings, the endothelial layer was mechanically removed by gently rubbing the luminal surface of the aortic ring back and forth several times with plastic tubing. The aorta rings were stretched, maintained at the optimal tension of 2 g, and allowed to equilibrate for 1 h. Endothelial integrity or functional removal was verified by the presence or absence of the relaxant response to acetylcholine (1 μM) of the phenylephrine (1 μM) contracted vessels. Less than a 10% relaxation was considered evidence that the aortic rings were functionally denuded of the endothelium, whereas the ability of acetylcholine to induce more than 90% relaxation was considered evidence that endothelium was intact. After assessing the presence of functional endothelium, vascular tissues were allowed to regenerate for 1 h before the start of an experiment [40].

In the first series of experiments, the effects of the studied compounds on vascular tension were defined. Aortic rings with intact or denuded endothelium were contracted with phenylephrine (1 μM) to obtain a maximal response. Once the maximal response had been obtained, the aortic rings were exposed to cumulative doses of the studied compounds, and the responses were recorded. 

To define the mechanisms by which the compound PSB-KD-107 relaxes a rat aorta, another series of experiments were done in aortic rings. The rings were exposed to various modulating agents such as L-NAME (100 μM), indomethacin (10 μM), or TEA (1 mM) for 20 min, and then vascular relaxation was carried out by cumulative additions of KD-107 at the plateau of the phenylephrine-induced contractions.

As L-NAME enhanced phenylephrine-induced contraction, the rings exposed to L-NAME were precontracted with 0.6 μM (instead of 1 μM) of phenylephrine to induce a magnitude of contraction similar to the one found in the rings not exposed to L-NAME [40].

#### 4.3.2. Influence of PSB-KD-107 and PSB-CB-92 on the Guinea Pig Ileum Contraction Induced by Carbachol

A 15 cm ileum segment was excised from the small intestine of male guinea pigs and immersed into a Krebs solution (NaCl 120 mM, KCl 5.6 mM, MgCl_2_ 2.2 mM, CaCl_2_ 2.4 mM, NaHCO_3_ 19 mM, glucose 10 mM). After the first 5 cm closest to the ileocaecal junction had been discarded, 2 cm-long fragments were cut. Each segment of the ileum was placed in the 20 mL chamber of tissue organ bath system (Tissue Organ Bath System—750 TOBS, DMT, Hinnerup, Denmark) filled with the constantly oxygenated (O_2_/CO_2_, 19:1) Krebs solution (37 °C, pH 7.4) and stretched by means of closing clips between the metal rod and the force-displacement transducer. The preparations, washed every 15 min with fresh Krebs solution, were allowed to stabilize in the organ baths for 1 h under a resting tension of 0.5 g. After the equilibration period, a cumulative concentration-response curve was constructed to carbachol (3 nM–3 μM) [41]. Following the first curve, tissues were incubated for 15 min with one concentration of the tested compound, and the next cumulative concentration curve to the agonist was obtained. Only one concentration of the potential antagonist was tested in each piece of tissue. 

#### 4.3.3. In Vitro Aggregation Test

In vitro aggregation tests using freshly collected whole rat blood were conducted with a Multiplate platelet function analyzer (Roche Diagnostic); the five-channel aggregometer was based on the measurements of electric impedance. Blood was drawn from rats’ carotid arteries with hirudin blood tube (Roche Diagnostic). Anticoagulated blood (300 μL) was mixed with 300 μL of the prewarmed isotonic saline solution containing studied compound or vehicle (deionized water or DMSO 0.1%) and preincubated for 3 min at 37 °C with continuous stirring. Aggregation was induced by adding collagen (Hyphen-Biomed, France) at the final concentration of 1.6 µg/mL. Activated platelet function aggregation process was recorded for 6 min. The Multiplate software analyzed the area under the curve (AUC) of the clotting process for each measurement and calculated the mean values.

#### 4.3.4. Antioxidant Assays

Antioxidant properties of investigated compounds were tested in vitro in two different assays: the 2,2-diphenyl-1-picrylhydrazyl (DPPH) assay [42,43,44] and the FeCl_3_ activity reduction assay (FRAP, ferric reducing antioxidant power) [43,44]. The tested compounds or ascorbic acid were dissolved in DMSO or in water (respectively) at 10^−2^ M and then diluted with ethanol.

##### DPPH Assay

A solution of 8 mg of DPPH (1,1-diphenyl-2-picrylhydrazyl; Sigma-Aldrich, Poland) in 20 mL of methanol was stored in the dark for 2 h. Solutions of selected phenols were prepared in methanol at the concentration of 1 mg/mL. 

Measurements were performed according to Brand-Williams et al. [45], with some modifications. In six test tubes, 0, 20, 40, 60, 80, and 100 µL of phenol samples were mixed with methanol to a final volume of 100 µL. Next, 2 mL of methanol and 0.25 mL of the prepared DPPH solution were added to each tube. The mixtures were vortexed and allowed to stand for 20 min in the dark at room temperature (25 °C). Absorbance was measured at 517 nm against methanol as a blank. 

##### FRAP Assay

FeSO_4_ × 7H_2_O used for calibration curve construction was dissolved in methanol and further diluted with water to obtain concentrations of 0.0125, 0.025, 0.05, 0.10, 0.15, 0.20, and 0.30 mg/mL. From the phenol samples, 1 mg/mL methanol solutions were prepared and diluted with water to obtain seven concentrations: 0.1, 0.08, 0.06, 0.04, 0.03, 0.02, and 0.01 mg/mL.

The assay was performed according to Benzie and Strain [46] with some modifications. The FRAP working solution was prepared prior to the start of the analysis: 0.3 mol acetate buffer (pH 3.6), 0.01 mol TPTZ (2,4,6-tripyridyl-s-triazine; Sigma-Aldrich) in 0.04 mol HCl (POCh, Gliwice, Poland), and 0.02M FeCl_3_×6H_2_O in water (iron (III) chloride hexahydrate; Chempur, Poland) were mixed in a volumetric ratio of 10:1:1 and protected from light. Next, 20 µL of the phenol solutions or FeSO_4_ × 7H_2_O solutions were mixed with 2.25 mL of the FRAP working solution and 180 µL of water. The obtained mixtures were incubated at 37 °C for 30 min, and their absorbance was measured at 593 nm. Deionized water with FRAP solution was used as a blank. 

### 4.4. In Vivo Studies

#### 4.4.1. Influence of PSB-KD-107 and PSB-CB-92 on Blood Pressure in Rats 

Rats were anesthetized with thiopental (70 mg/kg) via i.p. injection. The left carotid artery was cannulated with polyethylene tubing filled with heparin solution in saline to facilitate pressure measurements using a PowerLab Apparatus (ADInstruments, Sydney, Australia). After a 20 min stabilization period, the test compounds were administered i.p. at a dose of 10 mg/kg b.w in a constant volume of 1 mL/kg b.w. Blood pressure was recorded immediately prior to the administration and 5, 10, 20, 30, 40, 50, and 60 min after the test compound administration [47].

The effect of test substances on blood pressure after eight administrations was also determined. Rats were administered the test compound (10 mg/kg i.p.) or the vehicle (1% Tween 1 mL/kg i.p.) for 7 days, and then on the eighth day, the blood pressure was measured immediately before the compound administration and 5, 10, 20, 30, 40, 50, and 60 min thereafter.

#### 4.4.2. The Effect of PSB-KD-107 and PSB-CB-92 on Normal Electrocardiogram

In vivo electrocardiographic measurements were carried out using an ASPEL ASCARD B5 apparatus, standard lead II, and a paper speed of 50 mm/s. The study was performed after a single administration of the test compounds (10 mg/kg i.p.) and after eight once-daily administrations at the same dose. The ECG was recorded just before the administration of the compound and 5, 10, 20, 30, 40, 50, 60 min after administration. Then the following parameters were calculated and analyzed: duration of PQ, QT, QRS, RR intervals, and heart rate. Corrected QT interval (QTc) was calculated from Bazett’s formula: QTc = QT/√RR [48]. 

### 4.5. Data Analysis

All results are expressed as mean ± SEM or Δ/2, where Δ is a width of the 95% confidence interval (CI). Concentration-response curves were constructed based on the responses to cumulative concentrations of the studied compounds and analyzed by non-linear curve fitting using GraphPad Prism 6.0 (GraphPad Software Inc., San Diego, CA, USA). Relaxations were expressed as a percentage of inhibition of the maximal tension obtained with the contractile agent (E_max_ = 100%). The negative logarithm of the drug concentration that produced half of the maximum relaxation (pIC_50_) and the maximum response (E_max_) were calculated.

Statistically significant differences between groups were calculated using either one-way analysis of variance (ANOVA) and the post hoc Dunnett’s test and two-way ANOVA with interactions and the post hoc Šidák’s multiple comparison test, where appropriate. The criterion for significance was set at *p* < 0.05.

## 5. Conclusions

Our research clearly shows that PSB-KD-107, a GPR18 agonist, is selective versus other cannabinoid receptors and dilates blood vessels. This is probably related to its activity at this receptor, as vasodilation is reduced in the presence of PSB-CB-92, a GPR18 antagonist. Additionally, our findings indirectly confirm the presence of GPR18 in the blood vessels. The mechanism of the vasorelaxant activity of PSB-KD-107 is related to endothelial NO generation, however, we cannot exclude additional nitric oxide-independent mechanisms or the direct influence on K^+^ channels. Although PSB-KD-107 may affect blood pressure and heart function after a single administration, this effect was no longer observed after repeated (eight times once daily) administrations. PSB-KD-107 does not affect platelet aggregation—an important feature considering the safety of its administration. This compound also shows a significant antioxidant effect and further studies of its antioxidant activity in vivo are justified. Both compounds tested in this study—PSB-KD-107 and PSB-CB-92, are excellent tools for research on GPR18 functions.

## Figures and Tables

**Figure 1 pharmaceuticals-14-00799-f001:**
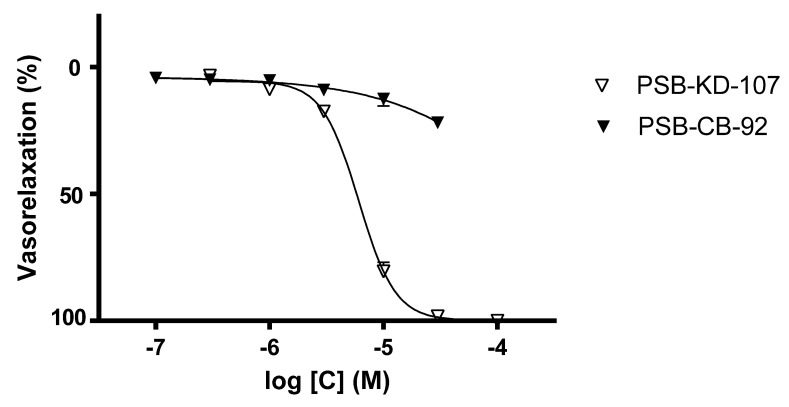
Concentration-response curves showing the vasorelaxant effects of PSB-KD-107 and PSB-CB-92 on endothelium-intact rat aortic rings, precontracted with phenylephrine (1 μM). Data are expressed as mean ± SEM of 3 experiments and represent a percentage of relaxation in phenylephrine-induced contraction.

**Figure 2 pharmaceuticals-14-00799-f002:**
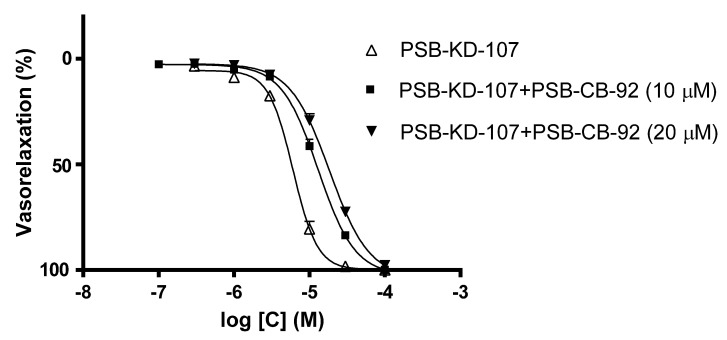
Concentration-response curves showing the vasorelaxant effect of (Δ) PSB-KD-107, (■) PSB-KD-107 with PSB-CB-92 (10 µM) and (▼) PSB-KD-107 with PSB-CB-92 (20 µM) in endothelium-intact rat aortic rings, precontracted with phenylephrine (1 μM). Data are expressed as mean ± SEM of 3–4 experiments and represent a percentage of relaxation in phenylephrine-induced contraction.

**Figure 3 pharmaceuticals-14-00799-f003:**
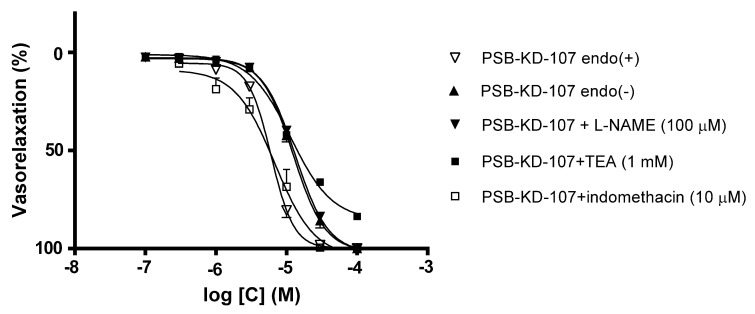
Effects of endothelium denudation (▲), L-NAME (▼,100 µM), TEA (■, 1 mM), and indomethacin (□, 10 µM) on PSB-KD-107 induced relaxation in phenylephrine-precontracted aortic rings. Data are expressed as means ± SEM of 3–4 experiments and represent a percentage of relaxation in phenylephrine-induced contraction.

**Figure 4 pharmaceuticals-14-00799-f004:**
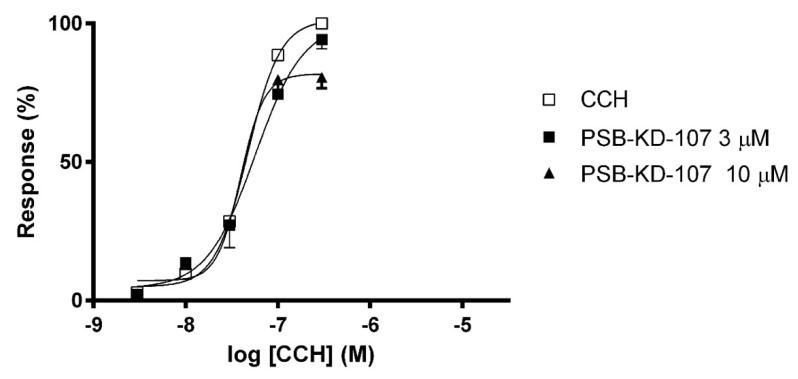
Concentration-response curves to carbachol (CCH) in the guinea-pig ileum in the absence (□) or presence of PSB-KD-107 (■ 3 μM, ▲10 μM). Results are expressed as percentage of the maximal response to CCH in the first concentration-response curve. Each point represents the mean ± SEM (*n* = 3).

**Figure 5 pharmaceuticals-14-00799-f005:**
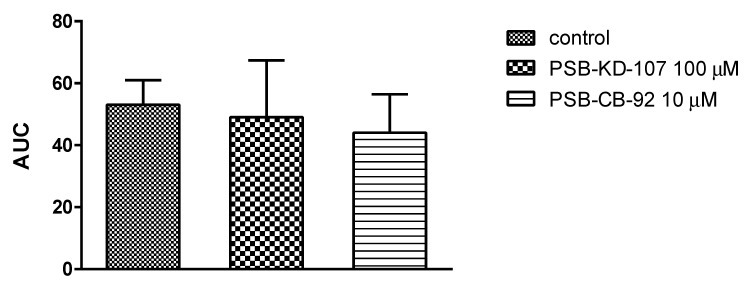
Effects of the studied compounds on in vitro whole rat blood aggregation induced by collagen (1.6 µg/mL). Results are expressed as mean + Δ/2, where Δ is a width of the 95% confidence interval (CI); *n* = 3–6; Statistical analysis: one-way ANOVA; AUC: area under curve.

**Figure 6 pharmaceuticals-14-00799-f006:**
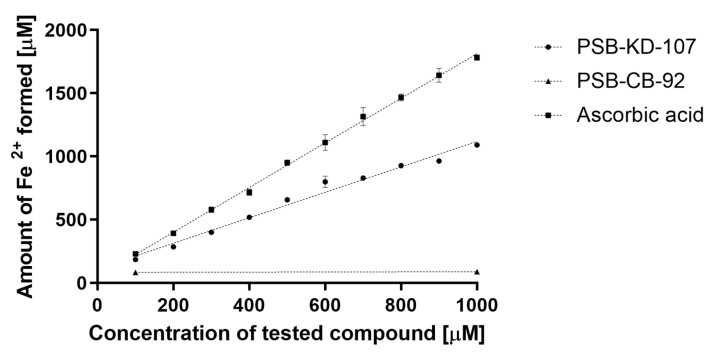
Antioxidant activities (FRAP assay) of PSB-KD-107 and PSB-CB-92 compared to ascorbic acid expressed as the amount of reduced iron ions (mean ± SEM). Linear regression lines are displayed.

**Figure 7 pharmaceuticals-14-00799-f007:**
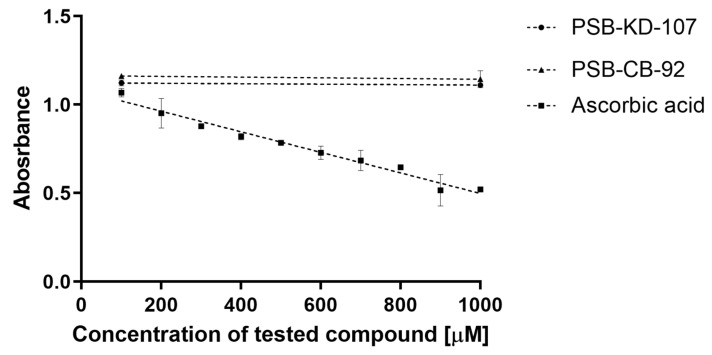
Antioxidant activities (DPPH assay) of PSB-KD-107 and PSB-CB-92 compared to ascorbic acid expressed as absorbance value (mean ± SEM).

**Figure 8 pharmaceuticals-14-00799-f008:**
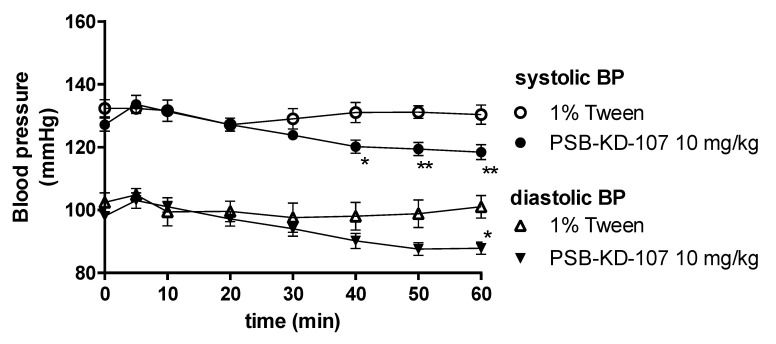
Effect of PSB-KD-107 on blood pressure after a single administration (10 mg/kg i.p.). Results are presented as mean ± SEM, *n* = 5. Statistical analysis: two-way ANOVA with interactions, post hoc Šidák’s multiple comparisons test, * *p* < 0.05; ** *p* < 0.01 vs. vehicle group.

**Figure 9 pharmaceuticals-14-00799-f009:**
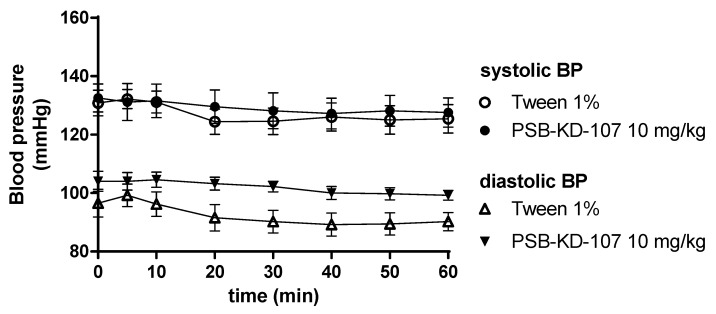
Effect of PSB-KD-107 on blood pressure after eight days of once-daily administrations (10 mg/kg i.p.). Results are presented as mean ± SEM, *n* = 6. Statistical analysis: two-way ANOVA, with interactions.

**Figure 10 pharmaceuticals-14-00799-f010:**
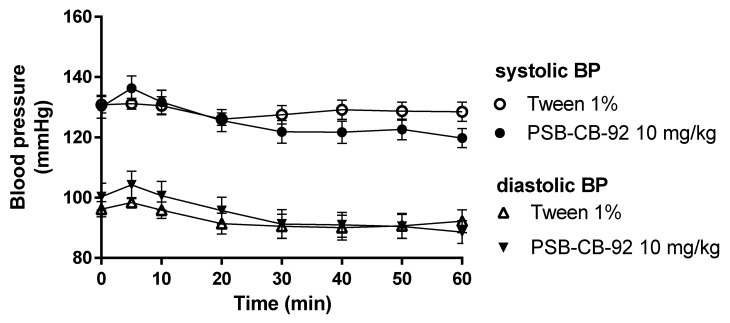
Effect of PSB-CB-92 on blood pressure after a single administration (10 mg/kg i.p.). Results are presented as mean ± SEM, *n* = 6. Statistical analysis: two-way ANOVA with interactions.

**Figure 11 pharmaceuticals-14-00799-f011:**
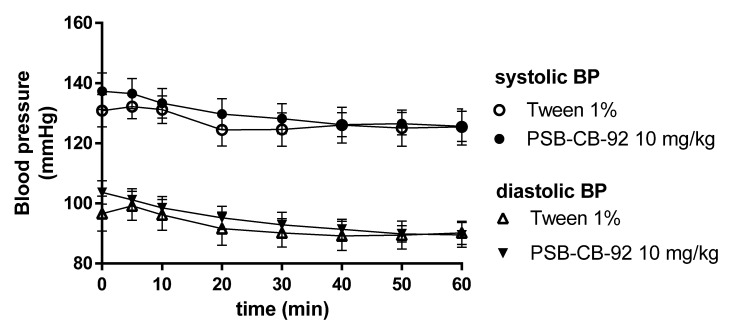
Effect of PSB-CB-92 on blood pressure after eight days of once-daily administrations (10 mg/kg i.p.). Results are presented as mean ± SEM, *n* = 6. Statistical analysis: two-way ANOVA with interactions.

**Figure 12 pharmaceuticals-14-00799-f012:**
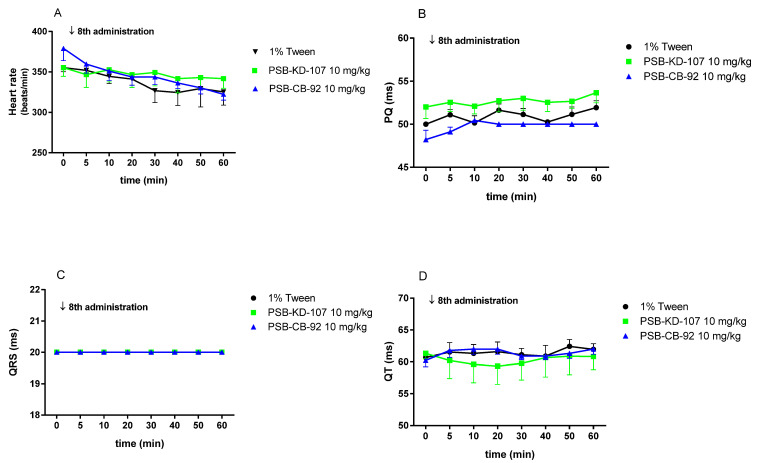
Effect of compounds PSB-KD-107, PSB-CB-92 (10 mg/kg i.p.) and the vehicle (1% Tween 1 mL/kg i.p.) on the heart rate (**A**), PQ interval (**B**), QRS complex (**C**), and QT interval (**D**), after eight once-daily administrations. Results are presented as mean ± SEM, *n* = 5. Statistical analysis: two-way ANOVA with interactions, post hoc: Šidák’s multiple comparisons test.

**Table 1 pharmaceuticals-14-00799-t001:** Pharmacological parameters for the concentration-relaxation curves of PSB-KD-107 and PSB-CB-92 under control conditions and after different pretreatments in rat thoracic aorta precontracted with phenylephrine.

Compound	Pretreatment	pIC_50_	Emax (%)
PSB-KD-107	none	5.217 ± 0.020	100.0 ± 0.0
	PSB-CB-92 (10 µM)	4.884 ± 0.020 ****	100.0 ± 0.0
	PSB-CB-92 (20 µM)	4.735 ± 0.023 ****	97.8 ± 0.8
	denudation	4.904 ± 0.024 ****	100.0 ± 0.0
	L-NAME (100 µM)	4.873 ± 0.015 ****	100.0 ± 0.0
	Indomethacin (10 µM)	5.165 ± 0.075	100.0 ± 0.0
	TEA (1 mM)	4.940 ± 0.030 ****	83.5 ± 1.5
PSB-CB-92	none	none	21.8 ± 1.60

Data are mean ± SEM from 3–4 experiments. Significant difference **** *p* < 0.0001 between the control and treatment group is indicated (one-way ANOVA, Dunnett’s post hoc test).

**Table 2 pharmaceuticals-14-00799-t002:** Effects of PSB-KD-107 and PSB-CB-92 on the rat ECG recording after a single administration (10 mg/kg i.p.).

Parameter		Time (min)							
		0	5	10	20	30	40	50	60
PSB-KD-107 10 mg/kg	Heart rate (beats/min)	334.0 ± 73.2	330.0 ± 77.2	331.3 ± 75.8	328.7 ± 78.6	311.3 ± 67.6	304.3 ± 60.2 *	316.0 ± 46.4	339.0 ± 27.4
	PQ (ms)	52.2 ± 1.7	51.1 ± 1.7	51.3 ± 1.0	50.9 ± 1.3	52.2 ± 1.7	52.6 ± 0.7	52.7 ± 0.8	52.9 ± 2.0
	QRS (ms)	20.0 ± 0.0	20.0 ± 0.0	20.0 ± 0.0	20.0 ± 0.0	20.0 ± 0.0	20.0 ± 0.0	20.0 ± 0.0	20.0 ± 0.0
	QT (ms)	63.3 ± 5.1	65.5 ± 3.4	65.5 ± 6.1	65.3 ± 3.7	64.4 ± 4.5	66.7 ± 2.9 **	65.5 ± 3.4	64.4 ± 1.7
	QT_c_ (ms)	132.8 ± 21.7	132.4 ± 24.4	129.7 ± 25.1	127.5 ± 25.6	131.1 ± 25.1	132.2± 24.5	132.9 ± 26.5	134.6 ± 19.6
PSB-CB-92 10 mg/kg	Heart rate (beats/min)	393.0 ± 27.4	376.7 ± 40.8 ***	385.7 ± 34.3	385.7 ± 34.3	384.7 ± 29.0	393.0 ± 27.4	393.0 ± 27.4	393.0 ±27.4
	PQ (ms)	48.9 ± 6.1	48.9 ± 4.5	49.8 ± 7.9	51.1 ± 7.4	48.9 ± 6.1	48.9 ± 6.1	51.1 ± 4.5	50.0 ± 8.8
	QRS (ms)	22.7 ± 5.4	23.6 ± 5.1	22.7 ± 5.4	22.7 ± 5.4	22.7 ± 5.4	22.7 ± 5.4	22.7 ± 5.4	22.7 ± 5.4
	QT (ms)	62.2 ± 3.4	64.4 ± 6.7	63.3 ± 5.1	63.3 ± 2.9	62.2 ± 3.4	62.2 ±3.4	63.3 ± 5.1	63.3 ± 2.9
	QT_c_ (ms)	159.1 ± 6.9	162.4 ± 13.4	159.2 ± 8.7	159.4 ± 1.0	157.7 ± 5.6	157.8 ± 6.4	160.0 ± 8.8	160.0 ± 8.0

Results are presented as means ± Δ/2, where Δ is a width of the 95% confidence interval (CI), *n* = 5. Statistical analysis: one-way RM ANOVA, post hoc Dunnett’s multiple comparisons test, * *p* < 0.05; ** *p* < 0.01, *** *p* < 0.001 vs. initial values.

## Data Availability

Data is contained within the article.

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
