# Peer review of "The GPR18 Agonist PSB-KD-107 Exerts Endothelium-Dependent Vasorelaxant Effects"

_pharmaceuticals, 2021, doi:10.3390/ph14080799_

Round 1

Reviewer 1 Report

The manuscript by KotaÅ„ska M. et al. entitled "The GPR18 agonist PSB-KD-107 exerts endothelium-dependent vasorelaxant effects" presents studies which evaluate the the effects of selective GPR18 receptor agonist – PSB-KD-107, as well as antagonist – PSB-CB-92, on blood pressure, electrocardiogram, vascular dilatation and their anti-oxidative potential. The results are novel and noteworthy. Manuscript is generally very well organized and very well-written. However, it suffers from a few minor problems that are listed below.

  1. Authors are inconsequent in the use of abbreviations. At first use the abbreviation should be explained and then used consistently throughout the manuscript.
  2. L. 84 is „rolesand” should be „roles and”
  3. L. 105 is „…in in vitro and in vivo.” should be “…in in vitro and in vivo research/studies”.
  4. L. 445 is „…with 95%O2/5%CO2and…” should be „…with 95%O2/5%CO2 and…”
  5. L. 480 is „…baths for 1h under… should be „…baths for 1 h under…”
  6. It seems to me that the bold names of the GPR18 receptor ligands were unnecessarily used.
  7. Materials and Methods

Animals subsection:

  1. Please provide the total number of rats used in the study.
  2. Were animals randomly selected to groups? Were behavioral experiments blinded? If yes, please explain how experiments were randomized and blinded. If not explain why they were not blinded and/or randomized.
  3. Please enter the breeder from whom the animals were purchased.
  4. Please state in what conditions (temperature, humidity) mice were kept and at what hours (time of day) experiments were carried out?

4.3.2. Influence of PSB-KD-107 and PSB-CB-92 on the guinea-pig ileum contraction induced by carbachol subsection:

  1. The Authors state that these studies were conducted on a 15 cm ileum segment excised from the small intestine of male guinea-pigs. However, in the Animals subsection, they did not provide any information about them.

Reviewer 2 Report

The manuscript by KotaÅ„ska et al addresses an important area of investigation pertaining to the novel cannabinoid receptor GPR18. Here they focus on the cardiovascular effects of the receptor ligands. The research design and the data presented is sound, and the authors do an excellent job at succinctly explaining the results. A few additional experiments, and a more detail on how these effects compare to the well-researched CB1R could greatly help strengthening the manuscript. 

1) I am not very surprised that you do not see prominent effects with the antagonist in your study. However, apart from one figure, you have not shown that the effect of the agonist (KD-107) is lost after pretreatment with the antagonist. Considering that there are not a lot of functional studies on GPR18 in blood vessels using these ligands, it is imperative that the specificity is confirmed by employing antagonists, or better yet using knock out models. Even in vitro experiments could also greatly help in strengthening some of your data. 

2) The use of the antagonist in figure 2 shows a significant, yet not a very prominent shift in the dose response curve. What is the range of percentage reduction in response you observed in the presence of the antagonist? Could you have perhaps employed a higher concentration of the antagonist to achieve a stronger reduction in response?

3) There needs to be some expression data shown to strengthen the findings. This could be in the form of immunoblotting or PCR. I understand that these ligands are claimed to be selective to GPR18 and not to the other cannabinoid and non-cannabinoid receptors. But that needs to be confirmed in your model. 

4) In view of point 4, there are some similarities in cardiovascular effects to that of the CB1R (vasodilation, not pronounced effects on heart rate in normotensive rats). This makes it imperative to show that these are all CB1R-independent effects.

5) How does GPR18 compare with the cardiovascular effects observed with CB1R agonists? CB1R has been shown to exhibit a triphasic effect, and its hypotensive effects are more pronounced in hypertensive models than in normotensive models (Haspula and Clark., 2020). Since CB1R has been more extensively researched, it becomes important to discuss your findings in lieu of previously established cardiovascular data on the CB1R. 

6) Are the in vivo effects gender specific? Do you observe similar effects in female mice?

7) How about changes in body weight, or other metabolic parameters?

8) Is the effect on heart mediated via the receptors located in cardiac tissue or is this an effect observed due to interaction with its receptors in the RVLM? Connected to this, can the ligands penetrate the BBB?

Reviewer 3 Report

This study describes the vasodilatator effect of PSB-KD-107, a GPR18 agonist, on blood vessels. An exhaustive presentation, with many well-explained details. An extremely complex method with studies done on rats. From the point of view of the text, I did not discover any significant grammatical errors. I believe that the possible cardiovascular side effects of PSB-KD-107 should be further studied and further studies of its antioxidant 
activity. A complex presentation, which will bring a degree of novelty and which I consider to be published in its current form

Author Response

Reviewer 3

This study describes the vasodilatator effect of PSB-KD-107, a GPR18 agonist, on blood vessels. An exhaustive presentation, with many well-explained details. An extremely complex method with studies done on rats. From the point of view of the text, I did not discover any significant grammatical errors. I believe that the possible cardiovascular side effects of PSB-KD-107 should be further studied and further studies of its antioxidant 
activity. A complex presentation, which will bring a degree of novelty and which I consider to be published in its current form

Authors: We kindly thank you for your positive feedback.

Round 2

Reviewer 2 Report

The authors have answered all my questions/concerns